# Effects of *Cysticercus cellulosae* Excretory–Secretory Antigens on the TGF-β Signaling Pathway and Th17 Cell Differentiation in Piglets, a Proteomic Analysis

**DOI:** 10.3390/microorganisms11030601

**Published:** 2023-02-27

**Authors:** Wei He, Qianqian Mu, Lizhu Li, Xiaoqing Sun, Xianmin Fan, Fengjiao Yang, Meichen Liu, Biying Zhou

**Affiliations:** Department of Parasitology, Zunyi Medical University, Zunyi 563000, China

**Keywords:** excretory–secretory antigens, proteomics, TGF-β signaling pathway, Th17 cells differentiation

## Abstract

Excretory–secretory antigens (ESAs) of *Cysticercus cellulosae* can directly regulate the proliferation and differentiation of host T regulatory (Treg) cells, thus inhibiting host immune responses. However, previous studies have only focused on this phenomenon, and the molecular mechanisms behind the ways in which *C. cellulosae* ESAs regulate the differentiation of host Treg/Th17 cells have not been reported. We collected CD3^+^ T cells stimulated by *C. cellulosae* ESAs through magnetic bead sorting and used label-free quantification (LFQ) proteomics techniques to analyze the signaling pathways of *C. cellulosae* ESAs regulating Treg/Th17 cell differentiation. Through gene set enrichment analysis (GSEA), we found that *C. cellulosae* ESAs could upregulate the TGF-β signaling pathway and downregulate Th17 cell differentiation in piglet T cells. Interestingly, we also found that the IL-2/STAT5 signaling pathway also affects the downregulation of Th17 cell differentiation. *C. cellulosae* ESAs activate the TGF-β signaling pathway and the IL-2/STAT5 signaling pathway in host T cells to further regulate the differentiation of Treg/Th17 cells in order to evade host immune attack. This study lays the foundation for the subsequent verification of these pathways, and further clarifies the molecular mechanism of *C. cellulosae*-mediated immune evasion.

## 1. Introduction

*Taenia solium* (*T. solium*) cysticercosis/taeniasis (TSCT) has been classified as a zoonotic parasitic disease by the World Health Organization (WHO). The *T. solium* life cycle consists of four stages of development: eggs, six-hook larvae, metacestode (*Cysticercus cellulosae*) and adults (*T. Solium*), and both *C. cellulosae* and *T. solium* can cause disease. When pork containing *C. cellulosae* is eaten raw or semi-raw, the digestive juices of the small intestine stimulate the *C. cellulosae* to emerge from the cephalic nodes, which then further develop into *T. solium*, causing taeniasis, which can cause clinical manifestations of the gastrointestinal tract such as abdominal pain, diarrhea, or dyspepsia. People can become infected with cysticercosis if they mistakenly ingest the eggs of *T. solium*. The eggs are stimulated by digestive juices to hatch into *six-hook larvae*, which burrow into the intestinal wall allowing dissemination throughout the body via the bloodstream, leading to gradual development of *C. cellulosae*. This results in cysticercosis, where patients often have seizures and which can cause more serious damage than taeniasis. Patients often have seizures, accompanied by symptoms such as dizziness, nausea, vomiting, and disturbance of consciousness [1,2,3]. *T. solium* causes mutual infection between humans and pigs, leading to a vicious circle that poses significant risk to human public health safety and animal husbandry [4,5].

When helminth parasites live in the host, they can excrete or secrete some antigens to the host, which plays an important role in tissue penetration, immune invasion, and host–parasite interactions [6]. Excretory–secretory antigens (ESAs) can induce the immune regulation of host T lymphocytes by secretory means, which facilitates the survival of the worms in vivo because they can evade host immune attack [7,8]. For example, *Echinococcus multilocularis* and *Echinococcus granulosus* protoscolex ESAs have been separately co-cultured with naive mouse CD4^+^CD25^−^ T cells and they both induced the transformation of naive CD4^+^CD25^−^ T cells into CD4^+^CD25^+^ Tregs [9,10]. ESAs of the *E. multilocularis metacestode* have been shown to drive the differentiation of naïve T cells to Foxp3^+^ Tregs and the production of IL-10 in vitro [11]. In addition, this study showed that host TGF-β and TGF-β signaling pathways play a crucial role in the differentiation of Treg cells [11]. This suggests that tapeworm ESAs can induce host Treg cell differentiation resulting in negative immune modulation. *C. cellulosae* ESAs can be used as specific antigens for the diagnosis of cysticercosis, as well as new drug targets [12]. We also previously found that *C. cellulosae* ESAs can induce dysregulation of CD4^+^/CD8^+^ T lymphocyte ratios and disrupt T lymphocyte immune function. This is in addition to inducing host T lymphocyte differentiation toward Treg cells, thereby exerting an immune evasion effect [13].

The transforming growth factor-β (TGF-β) signaling pathway is extremely important for regulating mammalian T lymphocyte immune responses and is mainly dependent on Smad proteins to mediate signaling [14]. Type I receptor (TβR-I) and type II receptor (TβR-II) are the receptor proteins involved in the TGF-β signaling pathway [15,16]. TGF-β binds to TβR-I and TβR-II receptors to form complexes that trigger intracellular signaling. TβR-II phosphorylation activates TβR-I, while phosphorylated TβR-I can further activate the downstream Smad2/3 proteins, and phosphorylated R-Smad2 and R-Smad3 combine with Smad-4 to form a trimeric complex and are displaced to the nucleus, where they interact with retinoid-related orphan nuclear receptor-γt (ROR-γt), a specific transcription factor for Th17 cells, or fork-head box p3 (Foxp3), a specific marker for Tregs, to regulate gene transcription and ultimately regulate the differentiation of Th17 or Treg cells [17,18] (Figure 1).

Proteomics has become a powerful tool for high-throughput characterization of global protein expression by combining mass spectrometry and bio-informatics and has become a research hotspot in parasite studies [19,20,21]. In recent years, researchers have used proteomics technology to conduct extensive studies on parasite ESAs. Proteomics can be used to screen proteins in helminth parasite ESAs that may play a key role during infection and suppress host immune responses, and that may also have potential as diagnostic antigens for worm diseases and vaccine candidate molecules. [22,23,24,25]. Since the mechanisms by which *C. cellulosae* ESAs induce immune evasion in piglet T cells are complex and not yet fully elucidated, we used label-free quantification (LFQ) proteomics techniques to further explore the molecular mechanisms by which *C. cellulosae* ESAs induce differentiation of piglet Treg/Th17 cells through the TGF-β signaling pathway and Th17 cell differentiation. This study has laid the foundation for elucidating the mechanisms by which ESAs regulate Treg/Th17 cell production in piglets.

## 2. Materials and Methods

### 2.1. Piglets and C. cellulosae ESAs

Healthy piglets were fecally examined by microscopy prior to infection and were confirmed to be free of pathogens. *C. cellulosae* ESAs were prepared as previously described [13]. Briefly, *C. cellulosae* were collected from the muscle tissue of the model piglets with cysticercosis, and the collected *C. cellulosae* were washed five times in sterile saline and phosphate-buffered saline (PBS). The *C. cellulosae* were incubated at a density of 10 cysticerci/mL in RPMI 1640 medium containing 2% glucose, 100 μg/mL streptomycin, and 100 U/mL penicillin at 37 °C and 5% CO_2_ for 72 h. The culture supernatant was then collected, concentrated (hereafter referred to as the ESAs), and stored at −80 °C until further use.

### 2.2. Protein Extraction

Section 2.2 and Section 2.3 were repeated twice and CD3^+^ T cells from the *C. cellulosae* ESAs stimulation and control groups were collected for protein extraction. The lysis buffer (Merck Millipore, Darmstadt, Germany) was added to CD3^+^ T cells and lysed by ultrasound. The supernatant was collected by centrifugation at 13,400× *g* for 10 min at low temperature (4 °C) and the protein concentration was determined using the BCA kit (Beyotime Biotech, Shanghai, China).

### 2.3. The Isolation and Culture of Peripheral Blood Mononuclear Cells (PBMCs)

First, 5 mL of fresh anticoagulated blood was aseptically drawn from the anterior vena cava of healthy piglets, and the fresh blood was mixed with porcine peripheral blood lymphocyte sample diluent (TBD Sciences, Tianjin, China) at a ratio of 1:2. In a 15 mL sterile centrifuge tube, 5 mL of porcine peripheral blood lymphocyte isolate (TBD Sciences) was added. Next, 5 mL of the diluted blood sample was then slowly added along the wall of the tube to the surface of the isolate and centrifuged at 315× *g* for 30 min at room temperature. The middle white cloudy layer (i.e., PBMCs) was carefully collected. The cells were then suspended by adding 10 mL of washing solution, centrifuged at 315× *g* for 10 min at low temperature (4 °C) and washed twice. If erythrocytes were left in the sediment, the erythrocytes were lysed with erythrocyte lysis solution (Solarbio Sciences, Beijing, China). The cell precipitate was resuspended in RPMI 1640 medium containing 15% fetal bovine serum and 1% antibiotics (penicillin and streptomycin) (Solarbio Sciences) [26]. PBMC were inoculated in 6-well plates at 1 × 10^6^ cells/well. ESAs (50 μg/mL) and RPMI 1640 medium (blank control) were added. After incubation at 37 °C and 5% CO_2_ for 3 h, 2.5 μg/mL of phytohemagglutinin (PHA) (Solarbio Sciences) was added for induction. Incubation was continued for 48 h, and cells were collected for magnetic bead sorting.

### 2.4. Isolation of CD3^+^ T Cells by Magnetic Bead Sorting

PBMCs were placed in 1.5 mL centrifuge tubes, resuspended with RPMI 1640 medium, and the supernatant was discarded by centrifugation at 315× *g* for 10 min. Next, 400 μL of staining buffer and 2 μL of CD3 (-PE Cy7) antibody (BD Pharmingen, Franklin Lakes, NJ, USA) were added and incubated for 30 min at 4 °C in the dark. Staining buffer was added and cells were washed and centrifuged at 315× *g* for 10 min. Subsequently, 80 μL of staining buffer and 20 μL of Anti-Cy7 microbeads (Miltenyi Biotec, Bergisch Gladbach, Germany) were added and incubated for 15 min at 4 °C in the dark. Then, 2 mL of staining buffer was added and cells were washed by blowing and mixing. Cells were then centrifuged at 315× *g* for 10 min, and the cell precipitate was resuspended in 600 μL of staining buffer. The LS column was then placed on the MACS sorter before the cell suspension was added to the LS column. Following binding to the column, piglet CD3^+^ T cells were collected. A CD3 (-PE Cy7) antibody was used to detect the purity of CD3^+^ T cells, and the CD4 (-PE Cy7) antibody (BD Pharmingen) was used to measure the proportion of CD4^+^ T cells by flow cytometry.

### 2.5. Pancreatic Enzyme Digestion

Lysis was carried out by adding an equal amount of lysis solution to the protein sample. After heating at 95 °C for 10 min and returning to room temperature, trypsin (Promega, Madison, WI, USA) was added and digested overnight. Dithiothreitol (Sigma, St. Louis, MO, USA) was added to the sample to a concentration of 5 mmol/L and reduction reaction at 56 °C for 30 min; iodoacetamide (Sigma) was then added to a concentration of 11 mmol/L and incubated for 15 min at room temperature in the dark [27].

### 2.6. Database Search of Liquid Chromatography–Mass Spectrometry (LC–MS)

LFQ technology is a protein quantification technique that is not dependent on isotope labeling. Analysis of the hydrolyzed peptides of proteins by LC–MS and comparison of the signal intensities of the peptides in the samples to the corresponding proteins allows for relative quantification [28,29,30].

The secondary mass spectrum data for this experiment were searched using Maxquant (v1.6.15.0, https://www.maxquant.org/) (accessed on 20 May 2022). Species retrieval parameters included the database *Sus scrofa domestica*_9823_PR_20220314 (49,793 sequences). The reference proteome of *Sus scrofa* was downloaded from the UniProt database (V. 2022_04). The reverse library was added to the database to calculate the false positive rate (FDR) and the common contamination library was added to eliminate contaminating proteins. The length of the minimum amino acid residue allowed for the peptide was 7. The maximum number of modifications allowed for peptides was 5. In the “First Search” and “Primary Search,” the mass error of the primary precursor ion was set to 20 ppm and the mass error of the secondary fragment ion was 20 ppm. The FDR for protein identification was set at 1%. The differentially expressed proteins (DEPs) between the two groups of CD3^+^ T cells were screened based on LC–MS analysis. The ratio of the mean relative quantification values of each protein in multiple replicate samples was taken as the fold change (FC).

### 2.7. Bio-Informatics Analysis and Statistical Analysis

#### 2.7.1. Annotation and Enrichment Analysis of Gene Ontology (GO)

The identified proteins were analyzed for annotation using eggnog-mapper software (v2.0), which is based on the EggNOG database and extracts the GO ID number from each protein annotation result. The DEPs were enriched for GO functional annotation in three main categories to elucidate their biological roles in terms of the different perspectives deriving from biological processes, cellular components, and molecular functions. GO enrichment significance analysis of DEPs was performed using Fisher’s exact test and a *p* value < 0.05 was considered to be statistically significant.

#### 2.7.2. Enrichment of Signaling Pathways

The Kyoto Encyclopedia of Genes and Genomes (KEGG) (https://www.kegg.jp/) (accessed on 18 July 2022) can integrate currently known protein interaction network information. Based on the KEGG pathway database, the identified DEPs were subjected to BLAST matching (blastp, evalue ≤ 1 × 10^−4^), and for the BLAST alignment results of each sequence, the comparison results with the highest alignment score were selected for pathway annotation. The Fisher’s exact test was used to analyze the significance of pathway enrichment of DEPs, and a *p* value < 0.05 was considered significant.

#### 2.7.3. Analysis of Gene Set Enrichment (GSEA) (https://www.gsea-msigdb.org/gsea/index.jsp) (accessed on 6 September 2022)

In terms of gene set enrichment, GSEA can more easily include the impact of subtle but coordinated changes on biological pathways. The advantage of this analysis is that it can scan all of a signaling pathway at once, can intuitively understand the expression of all proteins in each signaling pathway and can visualize which core proteins affect the upregulation/downregulation of the relevant signaling pathway.

All proteins were arranged in order from high to low according to the level of change (i.e., ratio) of the comparison group, which was used to indicate the trend in protein expression between the two groups. Among these, ratio values less than 1 were treated as the negative reciprocal. Enrichment analysis of the KEGG pathway was performed using GSEA (v 4.0.3) software. *p* values less than 0.05 were considered to represent significant enrichment.

#### 2.7.4. Analysis of Protein–Protein Interaction (PPI)

The differential protein interactions were extracted according to a confidence score > 0.7 following comparison with the STRING (v.11.0) protein interactions network database. Subsequently, differential protein interaction networks were visualized using the R package (networkD3) tool.

#### 2.7.5. Flow Cytometry Data Analysis

Flow cytometry statistical data analysis was conducted using IBM SPSS Statistics software (v.26.0). Statistical data were plotted in GraphPad Prism (v.8.0.2). *** *p* ≤ 0.001 was considered the highest statistically significant.

## 3. Results

### 3.1. Analysis of the Purity of CD3^+^ T Cells and the Ratio of CD4^+^ T Cells

CD3^+^ T cells were isolated from C. cellulosae ESA-treated PBMCs and the amount of CD4^+^ T cells was detected. Before magnetic bead sorting, the percentage of CD3^+^ T cells in the control and ESAs groups were 53.11% and 52.26%, respectively (Figure 2A). After magnetic bead sorting, the percentage of CD3^+^ T cells in the control and ESAs groups were determined, through flow analysis, to be 97.73% and 97.61%, respectively (Figure 2B), and the number of CD4^+^ T cells in the ESAs group was significantly higher than that in the control group (Figure 2C).

### 3.2. Analysis of DEPs by GO Functional Annotation and KEGG Signaling Pathway Enrichment

Based on LC–MS analysis, we identified 2150 DEPs with a FC > 1.2 in *C. cellulosae* ESAs/Control, of which 953 were upregulated and 1197 were downregulated (Appendix A). These DEPs were classified by GO secondary annotation, and 342 proteins were found to be involved in cellular processes, 112 proteins as signaling molecules, and 106 proteins were involved in T cell immune processes (Figure 3A). Enrichment through the KEGG pathway revealed that these DEPs are mainly involved in biological responses, catalytic activity, and metabolic processes of T cells, such as lysosome, synaptic vesicle cycle, glycosaminoglycan degradation, glycosphingolipid biosynthesis-ganglio series, glutathione metabolism, and steroid biosynthesis, among others (Figure 3B).

### 3.3. C. cellulosae ESAs Could Activate the TGF-β Signaling Pathway in T Cells and Inhibit Th17 Differentiation

To investigate ESA signaling pathways that affect Treg/Th17 cell differentiation in piglets, based on the KEGG database, we further performed signaling pathway enrichment analysis of DEPs by GSEA, and found that the TGF-β signaling pathway (KEGG: map04350) was upregulated (Figure 4A). In the gene set of the TGF-β signaling pathway (Appendix A), these DEPs acted as core proteins to influence the upregulation of this pathway (Table 1). In addition, we also found that Th17 cell differentiation (KEGG: map04659) was downregulated (Figure 4B). In the gene concentration of Th17 cells differentiation pathway (Appendix A), these DEPs act as core proteins to influence the downregulation of this pathway (Table 2)

### 3.4. PPI Analysis of DEPs

According to GESA analysis, a total of 21 proteins were identified in the TGF-β signaling pathway and 56 proteins were identified in Th17 cell differentiation, we then performed PPI analysis of the identified 77 proteins. STAT3, JAK1, SMAD2, SMAD3, CD4, SLA-DRA1, and tyrosine-protein kinase were found to interact more closely with each other, forming a web of interconnections (Figure 5). It was suggested that these proteins may play an important role in the activation of these two pathways.

## 4. Discussion

Developing countries and associated regions still face great challenges in preventing and controlling TSCT [31,32]. *C. cellulosae* can secrete and excrete some products (e.g., ESAs) when it parasitizes its host. These products can modulate host production of Tregs, which exert immunosuppressive effects, thus promoting long term *C. cellulosae* infection in the host [13]. However, existing research has focused solely on this phenomenon, while the molecular mechanism behind *C. cellulosae* ESA-mediated regulation of Treg differentiation has not been fully elucidated. Therefore, we used the LFQ technique to investigate the molecular mechanisms behind this process in piglets through the TGF-β signaling pathway and Th17 cell differentiation.

Treg and Th17 cells play opposite roles in the immune responses to parasite infection, and disturbances in the balance between them exist in a variety of parasite infectious diseases and animal models. Maintaining Treg/Th17 cell homeostasis is important for the host’s immune status after parasitic infection [33,34]. It was found that *Fasciola hepatica* induced the production of Treg cells in mice through the secretion of IL-10 and TGF-β. The infection with *F. hepatica* not only suppressed the immune response to Th1 and Th2 types, but also to auto-antigen-specific Th1 and Th17 in auto-immune encephalomyelitis (EAE) [35]. The TGF-β signaling pathway can regulate Treg/Th17 cell differentiation by relying on its downstream signal protein, Smad. For example, studies have shown that when the *Echinococcus granulosus* cyst fluid stimulates the normal mouse splenocytes in vitro, transcriptional levels of *Foxp3* and *IL-17* were significantly increased, while *Smad2* and *Smad4* increased then decreased within 12 h of the stimulation of *E. granulosus* cyst fluid [36,37]. The transcriptional levels of *IL-17*, *TβRI*, *TβRII*, and *Smad2/3* in the liver tissue of a mouse model infected with *Echinococcus multilocularis* were significantly increased. However, in the case of prolonged infections, the percentage of CD4^+^CD25^+^Foxp3^+^ Treg cells and serum levels of TGF-β and IL-10 were also found to be increased [38]. This suggests that tapeworm infections could initiate the TGF-β/Smad signaling pathway, upregulate TβRI, TβRII, and Smad2/3, and in turn induce the host to secrete a large amount of IL-17A, thus stimulating predominantly Th17-mediated immune responses. In the case of prolonged infection, large amounts of TGF-β and IL-10 were secreted, which further induced the proliferation and differentiation of Tregs and enhanced their immunosuppressive function, thus damaging T cell-mediated immune responses to parasites and leading to persistent infectious inflammation.

In this study, through GSEA enrichment, we found that *C. cellulosae* ESAs activate the TGF-β signaling pathway in piglet T cells. A total of 21 proteins were identified in the TGF-β signaling pathway, of which nine were core proteins that affected the regulation of this pathway. This result is similar to previous data showing that *Heligmosomoides polygyrus* ESAs could increase the spleen CD4^+^Foxp3^+^ T cells ratio in C57BL/6 mice in vitro by activating the TGF-β/Smad signaling pathway [39]. One hypothesis is that the molecular mechanism of *C. cellulosae* ESA-mediated regulation of piglet Treg proliferation may be accomplished through the TGF-β/Smad signaling pathway. Zinc finger FYVE domain-containing protein 16 (ZFYVE16), also known as endofin, is a member of the FYVE structural domain protein family [40]. As a scaffold protein, endofin is involved in endosomal transport and promotes TGF-β signaling transduction [41,42]. Endofin interacts with the TβRI (ALK5) and Smad4, recruiting Smad4 into the TβRI complex and promoting the formation of the R–Smad–Smad4 heterodimeric complex, thus promoting TGF-β signaling transduction [43]. Yin found that the percentage of CD4^+^ Treg cells and the level of TGF-β1 in serum were increased significantly in the spleen of *E. granulosus* model mice. If the TGF-β/Smad signaling pathway was blocked with an ALK5 inhibitor, the percentage of CD4^+^ Treg cells and the level of Smad2/3 protein phosphorylation could be decreased, suggesting that ALK5 inhibition could block the TGF-β/Smad signaling pathway by decreasing Smad2/3 protein expression in *Eg* mouse model, and in turn inhibiting the differentiation of Treg cells [44].

We also found that *C. cellulosae* ESAs could inhibit the differentiation of piglet Th17 cells at 48 h by GSEA enrichment. In Th17 cell differentiation, a total of 56 proteins were identified, of which 25 core proteins affected the downregulation of this pathway. Among the core proteins enriched in Th17 cell differentiation, we found that both the TGF- β/Smad and IL-2/STAT5 signaling pathways affect differentiation of Th17 cells. The activation of these signaling pathways can promote the expression of Foxp3 transcription factor, which in turn inhibits the expression of the ROR-γt transcription factor, thereby suppressing the differentiation of Th17 cells [45,46]. The aromatic hydrocarbon receptor (AHR) is a ligand-activated transcription factor belonging to the bHLH-PAS family [47,48]. The activation of AhR plays distinct roles in cellular function, including immunomodulation and T cells differentiation. When an AhR ligand binds it can regulate the differentiation of CD4^+^ T cells into Th17 or Treg cells [49,50]. Related studies have found that AhR interacts with STAT5 and STAT1 to enhance their phosphorylation and inhibit the differentiation of Th17 cells by promoting STAT5 activity [51,52,53]. IL-2 is a pleiotropic chemokine and when it binds to the IL-2α receptor (IL-2Rα), it can induce the phosphorylation of the signaling molecule, STAT5, which forms a homodimer or tetramer structure. These structures translocate to the nucleus to combine with the non-coding sequence 2 of the conservative region of the Foxp3 gene enhancer, thus driving the proliferation and differentiation of Treg cells [46,54,55]. In addition, related studies have shown that the IL-2/STAT5 signaling pathway can also restrict T lymphocytes to differentiate into Th17 cells subsets [56]. Therefore, the IL-2/STAT5 signaling pathway can be regarded as the key regulation point for the Treg/Th17 cell differentiation induced by initial CD4^+^ T cells, promoting Treg proliferation but inhibiting Th17 cell differentiation [57,58]. In this study we have only investigated at the level of proteomics, and we will continue to verify these *C. cellulosae* ESA activated pathways individually using blockers (ALK5 inhibitors/Smad7), gene knockdown (IL-2^−/−^), and other techniques. Importantly, these data lay the foundation for further elucidating the mechanism of immune escape caused by *C. cellulosae*.

## 5. Conclusions

As previously mentioned, since the molecular mechanisms involved in the regulation of Treg/Th17 cells proliferation and differentiation by porcine *C. cellulosae* ESAs have not been explored, in this study, we used LFQ for GSEA enrichment, and found that the TGF-β/Smad signaling pathway was upregulated but that it inhibited the differentiation of Th17 cells, suggesting that *C. cellulosae* ESAs can regulate the proliferation of piglet Tregs and inhibit Th17 cell differentiation by activating TGF-β/Smad. Interestingly, we also found that the IL-2/STAT5 signaling pathway also inhibits differentiation of Th17 cells. This paves the way for subsequent studies.

## Figures and Tables

**Figure 1 microorganisms-11-00601-f001:**
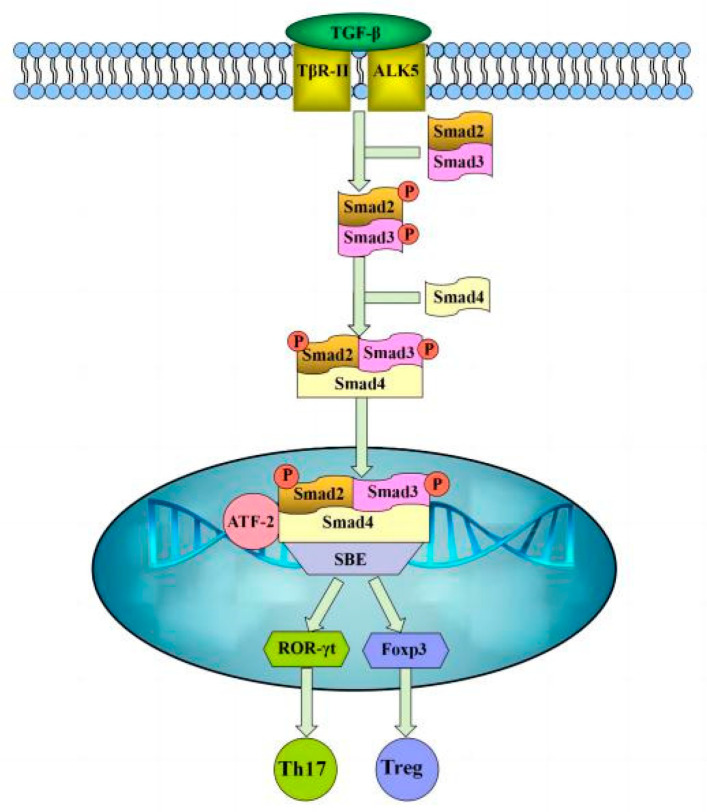
TGF-β signaling pathway.

**Figure 2 microorganisms-11-00601-f002:**
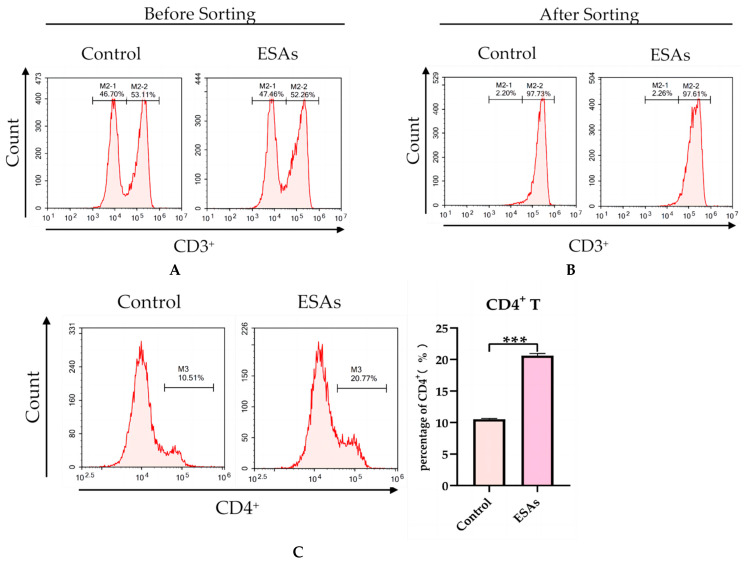
Analysis of the purity of CD3^+^ T cells and the ratio of CD4^+^ T cells. (**A**) The proportion of CD3^+^ T cells before magnetic bead sorting. (**B**) The proportion of CD3^+^ T cells after magnetic bead sorting. (**C**) Through flow cytometry analysis, the percentage of CD4^+^ T cells in the ESAs group was found to be significantly higher than that of the control group in terms of CD3^+^ T cells. *** *p* < 0.001.

**Figure 3 microorganisms-11-00601-f003:**
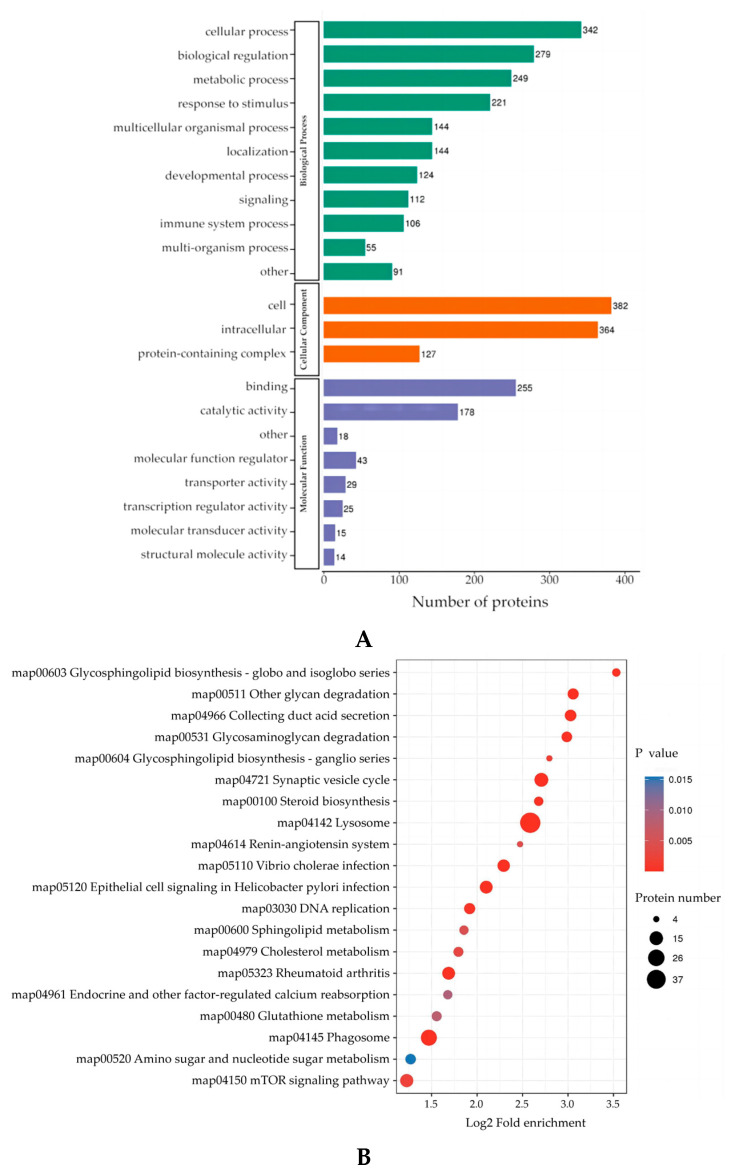
Analysis of DEPs by GO functional annotation and KEGG signaling pathway enrichment. (**A**) GO enrichment elaborates the differential protein functions in terms of biological processes, cellular components, and molecular functions. (**B**) The KEGG pathway enrichment bubble plot showed the functional classification and pathway from significant enrichment (*p* value < 0.05) of DEPs.

**Figure 4 microorganisms-11-00601-f004:**
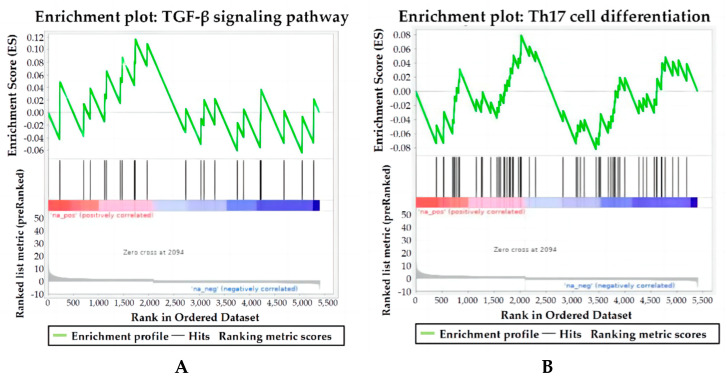
Pathway enrichment map of TGF-β signaling pathway and Th17 cell differentiation by GSEA. The graph is divided into three parts, the first part is the line graph of the protein enrichment score (ES). In the top panel, the x-axis depicts all proteins, the y-axis depicts the running ES corresponding to the protein in the pathway, the peaks refer to the ES of the pathway, and the protein before the peak is the core protein of the pathway. For the middle panel, the pathway enrichment map refers to the hit, which marks the proteins located in this pathway with lines, and the color bars below indicate the corresponding expression amount of the proteins. The bottom panel shows the distribution plot of the quantitative value ratio for all proteins. (**A**) The enrichment map of the TGF-β signaling pathway is an upregulated pathway. (**B**) The enrichment map of the pathway for Th17 cells differentiation is a downregulated pathway.

**Figure 5 microorganisms-11-00601-f005:**
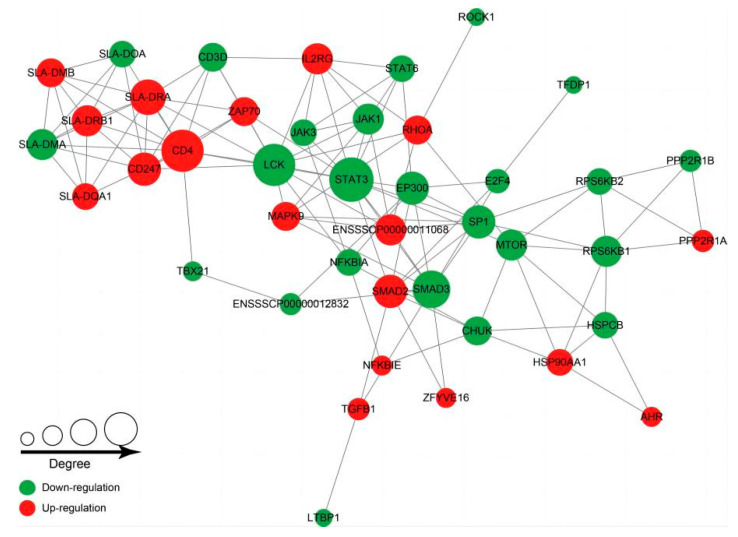
PPI analysis was carried out for the enriched DEPs of the TGF- β signaling pathway and Th17 cell differentiation. Circles indicate DEPs, green circles are downregulated proteins and red circles are upregulated proteins.

**Table 1 microorganisms-11-00601-t001:** Core proteins that promoted upregulation of the TGF-β signaling pathway.

Protein Accession	Protein Description	Gene Name	Rank Metric Score	Core Protein
F1RF24	Zinc finger FYVE domain-containing protein 16	ZFYVE16	2.586999893	Yes
A0A287BLR9	Receptor protein serine/threonine kinase	TGFβR1	1.463999987	Yes
F1RPR3	Mothers against decapentaplegic homolog	SMAD2	1.353999972	Yes
A0A5G2R017	S-phase kinase-associated protein 1	SKP1	1.218000054	Yes
A0A286ZT31	Mothers against decapentaplegic homolog	SMAD4	1.207999945	Yes
P07200	Transforming growth factor beta-1 proprotein	TGFβ1	1.125	Yes
A0A286ZMM6	Cullin 1	CUL1	1.11500001	Yes
P54612	Serine/threonine-protein phosphatase 2A 65 kDa regulatory subunit A alpha isoform	PPP2R1A	1.062999964	Yes
I3LVS7	Ras homolog family member A	RHOA	1.059999943	Yes

**Table 2 microorganisms-11-00601-t002:** Core proteins that inhibited the differentiation of Th17 cells.

Protein Accession	Protein Description	Gene Name	Rank Metric Score	Core Protein
Q85ZW4	MHC class II antigen OS = Sus scrofa	SLA-DRA1	1.889999986	Yes
K7GRS5	Stress-activated protein kinase JNK	MAPK8	1.657999992	Yes
I3LF82	Aryl hydrocarbon receptor	AHR	1.652999997	Yes
A0A287BLR9	Receptor protein serine/threonine kinase	TGFβR1	1.463999987	Yes
F6PX38	Ig-like domain-containing protein	SLA-DQB1	1.440999985	Yes
Q31072	HLA class II histocompatibility antigen, DRB1-4 beta chain isoform X2	LA-DRβ-d	1.429999948	Yes
Q7YQ94	MHC class II antigen	SLA-DQA1	1.406999946	Yes
F1RPR3	Mothers against decapentaplegic homolog	SMAD2	1.353999972	Yes
A0A286ZT31	Mothers against decapentaplegic homolog	SMAD4	1.207999945	Yes
O02705	Heat shock protein HSP 90-alpha	HSP90AA1	1.174000025	Yes
A0A287BF83	DUF1968 domain-containing protein	LOC102158035	1.167999983	Yes
P07200	Transforming growth factor beta-1 proprotein	TGFβ1	1.125	Yes
A0A287BD27	1-phosphatidylinositol 4,5-bisphosphate phosphodiesterase gamma	PLCG1	1.093999982	Yes
F1SLT4	T-cell surface antigen T4/Leu-3	CD4	1.085999966	Yes
A0A287A669	Interferon regulatory factor 4	IRF4	1.080000043	Yes
A5D9L3	MHC class II, DM beta	SLA-DMB	1.067000031	Yes
A0A5K1VIZ8	Interleukin-4 receptor subunit alpha	IL4R	1.057000041	Yes
F1RQU0	NFKB inhibitor epsilon	NFKBIE	1.047999978	Yes
A0A5G2R8A6	Serine/threonine-protein phosphatase	PPP3CB	1.046000004	Yes
F6PZ59	Cytokine receptor common subunit gamma	IL2RG	1.039000034	Yes
A0A287AYI3	Signal transducer and activator of transcription	STAT5B	1.034999967	Yes
A0A286ZPF5	Mitogen-activated protein kinase	MAPK1	1.018000007	Yes
F1S5Q3	Stress-activated protein kinase JNK	MAPK9	1.013000011	Yes
A0A287AQZ8	T-cell receptor T3 zeta chain	CD247	1.011000037	Yes
A0A5K1V762	Tyrosine-protein kinase	ZAP70	1.011000037	Yes

## Data Availability

The raw data presented in the study are deposited in the 4TU. Under the following https://doi.org/10.4121/21805041.v1 (accessed on 3 January 2023).

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
