# Peer review of "Effects of Cysticercus cellulosae Excretory–Secretory Antigens on the TGF-β Signaling Pathway and Th17 Cell Differentiation in Piglets, a Proteomic Analysis"

_microorganisms, 2023, doi:10.3390/microorganisms11030601_

Round 1

Reviewer 1 Report

The effects of Cysticercus cellulosae excretory-secretory antigens on TGFβ signalling pathway and Th17 cells differentiation in piglets T cells by proteomics 

Section: microorganisms  

General comments 

The results and discussion need to be improved.  

Introduction 
Line 94-97: Please, re-write this sentence. It is difficult to understand what you want to say.  

Line 104: There is a double space here.  

Material and Methods 

Line 109-110: What do you mean by the previous stage? 

Line 112: PBMCs means Peripheral Blood Mononuclear Cells. Please, clarify this.  

Line 126: What it means PHA?  

215: Which organism did you use for this analysis? I did not find Taenia solium or Cysticercus cellulosae in the STRING tool.  

Results 

It is essential to reorganize this section and explore the results more.  

Figure 2: I think this figure should go in supplementary data. 

Table 1: I think this table is unnecessary. No new data is added. Please, delete it or replace it with a more informative one—for example, the top 10 most up and down proteins.  

Line 273: Please, delete the dot.  

Line 334: Please, clarify what Result 3 means.  

Discussion 

It is essential to add more discussion and comparison with Taenia crassiceps.  

Lines 363-366: This should go in M&M.  

Lines: 368-379: This should go in Results.

Author Response

Dear Reviewer 1

I am very grateful to you for your precious comments on manuscript: microbiology-2171539 in the midst of their busy schedule, and I am also grateful for the opportunity to revise this manuscript, which make it more and more perfect. My manuscript has been revised by native speakers through the organization. According to the suggestions, I would like to report the revised contents to you:

Introduction

Re 1: Thanks for your suggestion, this sentence has been re-writed. (Line 92-95)

Re 2: Thanks for your attention, the double space has been deleted. (Line 101)

Material and Methods 

Re 1: I apologise for not presenting this section clearly and making it difficult for you to read, I have now rewritten this section clearly, please check. (Line 107-114)

Re 2: Thanks for your reminder, PBMCs means Peripheral Blood Mononuclear Cells.  (Line 115)

Re 3: Phytohaemagglutinin (PHA) can induce the proliferation of T cells through TCR and CD2 pathways. (Line 130)

Re 4: Thanks for your question, the organism we used was Sus scrofa (Line 169-170). This experiment was to investigate the signaling molecules in porcine T cells, therefore the species chosen was Sus scrofa.

Results

Re 1: Thanks for your suggestion, The reason why I put Figure 2 in the results was to prove that Cysticercus cellulosae ESAs could stimulate the number of CD4+ T cells in piglets (Figure 2C). This corresponds to the results of the previous study (References 25). 

Re 2: Thanks for your suggestion, I have put Table 1 on the supplementary data.

Re 3: Thanks for your reminder, the dot has been deleted. (Line 262)

Re 4: Thanks for your suggestion, I have re-described this sentence. (Line 327-328)

Discussion

Thanks for your suggestion. For this part, I have added content related to tapeworm regulation of Treg/Th17 cells. (Line 360-367)

Re 1: Thanks for your suggestion, contents of Lines 363-366 have moved to M&M. (Lines 202-206)

Re 2: Thanks for your suggestion, contents of Lines 368-379 have moved to Results.

Reviewer 2 Report

Dear Authors,

Title: Suggest to remove "T cells" from the title. 

Line 8 and 32: Do not need to mention the shorten form, C. cellulosae within the brackets as its standard in scientific witting. 

Line 9: Mentioning at the 1st instance, T regulatory cells (Treg) will be beneficial. 

Line 16: Suggest to revise, ....ESAs could upregulate TGF-β....

General comment: Genus name can be shorten after using the full name in the 1st instance. Please pay attention to italicising where it needed (Eg. in vivo). Extensive English language editing is required. 

Section 2.1: Authors have cited a reference for SAE production. However it will be beneficial to present a summary of the methods for this manuscript. Similar to how authors have presented the section 2.5.

 Section 2.1: Please specify what are the procedures taken to confirm that the animals are pathogens free. 

Line 231-133: This sentence is unclear. 

Discussion: Please discuss any other pathways that can regulate Th17 and Treg function (if there are any). Please discuss total number of proteins present in TGF-β signaling pathway and Th17 cells differentiation pathway and the number of protein found in the analysis (up regulated or down regulated or no change).

Author Response

Dear Reviewer 2

I am very grateful to you for your precious comments on manuscript: microbiology-2171539 in the midst of their busy schedule, and I am also grateful for the opportunity to revise this manuscript, which make it more and more perfect. My manuscript has been revised by native speakers through the organization. According to the suggestions, I would like to report the revised contents to you:

Re 1: Thanks for your suggestion, "T cells" has been deleted at the title

Re 2: Thanks for your reminder, C. cellulosae within the brackets has been deleted. (Lines 8 and 30)

Re 3: Thanks for your reminder, “T regulatory (Treg)” has been added in line 9

Re 4: Thanks for your suggestion, It has been modified according to your suggestions. (Lines 15) 

Material and Methods

Section 2.1: Thanks for your suggestion, I apologise for not presenting this section clearly and making it difficult for you to read, I have now rewritten this section clearly, please check. (Line 107-114)

Section 2.1: Thanks for your reminder, we performed faecal examination by microscopy to determine that the piglets were free of other parasitic infections. (Line 104-105)

Re 5: Thanks for your suggestion, I have re-described this sentence. (Line 224-225)

Re 6: Thanks for your suggestion, for this part, I have added content related to tapeworm regulation of Treg/Th17 cells. (Line 360-367)

Re 7: Thanks for your suggestion, In the Discussion section, I have added the total amount of protein and the number of core proteins identified for the TGF-β signaling pathway (upregulated) (Line 376-378), as has the Th17 cell differentiation pathway. (Line 397-398)

Reviewer 3 Report

Authors described in the current manuscript study on effect of Cysticersus cellulosae excretory-secretory (ES) antigens on TGF-β signaling pathway and Th17 cell differentiation in piglets T cell using label free quantitative approach. Considering that Taenia solium is one of the most important zoonotic parasites, the authors have addressed the important issue of the influence of the ES antigen on the immune system of the main intermediate host of this parasite. The study is valuable, however, some sections of the manuscript need modification and/or explanation.

I have the following specific comments that should be clarified, corrected or considered:

§ I highly recommend checking and correcting the article for the correctness of the English language with the help of a native speaker preferably from the biological field.

§  I suggest to describe briefly the development cycle of Taenia solium in the introduction section. This may be helpful to the reader of the article.

§  I could not find in the Materials and Methods section how many biological replicates of isolated T lymphocytes of were analyzed using proteomics.

§  Line 179: Please provide details from which database you have downloaded the reference proteome of Sus scrofa domestica.

§  Please provide information with which software/tools the following analyses were performed: GO annotation (section 2.7.1), enrichment signaling pathways (2.7.2) , and GSEA (2.7.3).

§  I recommend submitting results from proteomic analyses to public depositories such as PRIDE.

Author Response

Dear Reviewer 3

I am very grateful to you for your precious comments on manuscript: microbiology-2171539 in the midst of their busy schedule, and I am also grateful for the opportunity to revise this manuscript, which make it more and more perfect. My manuscript has been revised by native speakers through the organization. According to the suggestions, I would like to report the revised contents to you:

Re 1: Thanks for your suggestion, In the introduction section, I have added a description of the developmental cycle of Taenia solium. (Line 29-43)

Re 2: In the Materials and Methods section, I have added relevant descriptions. (Line 149-150) (3 times biological replicates of isolated T lymphocytes of were analyzed using proteomics) 

Re 3: Thanks for your suggestion, The reference proteome of Sus scrofa was downloaded from the UniProt database (V. 2022_04). (Line 170-172)

Re 4: Thanks for your suggestion, I have added relevant descriptions. 

GO annotation: The identified proteins were analyzed for annotation using eggnog-mapper software (v2.0), which is based on the EggNOG database and extracts the GO ID number from each protein annotation result. (Line 184-186)

Enrichment of Signaling Pathways: Based on the KEGG pathway database, the identified differentially expressed proteins were subjected to BLAST matching (blastp, evalue ≤ 1e-4), and for the BLAST alignment results of each sequence, the comparison results with the highest alignment score were selected for pathway annotation. (Line 194-198)

GSEA: Enrichment analysis of the KEGG pathway was performed using GSEA (v 4.0.3) software. (Line 210-211)

Re 5: Thanks for your suggestion, I have submitted results from proteomic analyses to public depositories. (Line 441-442)

Round 2

Reviewer 1 Report

The effects of Cysticercus cellulosae excretory-secretory antigens on TGFβ signaling pathway and Th17 cells differentiation in piglets T cells by proteomics 

Section: microorganisms  

General comments on the first revision  

Introduction & Material and Methods 

Thank you very much for the modifications.  

Results 

I cannot find lines 327-328 in connection with response number 4 (Re 4: Thanks for your suggestion, I have re-described this sentence. (Line 327-328)). Could you please highlight this in the Ms?  

Discussion 

In connection with this comment: Thanks for your suggestion. For this part, I have added content related to the tapeworm regulation of Treg/Th17 cells. (Line 360-367). I think I can find this in lines 355-361. 

Try to write lines 359-360 again, it is redundant when you say liver tissue of mice and then mice model. As a suggestion, you can write something like this:  

The transcriptional levels of IL-17, TβRI, TβRII, and Smad2/3 in the liver tissue of mice model with Echinococcus multilocularis were significantly increased. However, in the case of prolonged infections, the percentage of CD4+CD25+Foxp3+ Treg cells and serum levels of TGF-β and IL-10 were founded also increased [36].  

New comments for this last version 

Please, include this paper in the Ms: Walsh KP, Brady MT, Finlay CM, Boon L, Mills KHG. Infection with a helminth parasite attenuates autoimmunity through TGF-β-mediated suppression of Th17 and Th1 responses. Journal of Immunology. 2009;183(3):1577–1586. 

Introduction & Material and Methods 

Line 30: I understand what you want to say but it is not completely correct. You should correct this part for a better understanding of eggs, six-hook larvae, metacestode (Cysticercus cellulosae) and adult (T. Solium), and both C. cellulosae and T. solium can cause disease. 

Line 46: Please, delete the host. It is redundant.  

Line 34: The term taeniasis solium is not normally used and could be weird to read. Please, change this term to taeniasis. Also, it is important to say that the clinical signs and symptoms are undesignable from other gastrointestinal infections, including T. saginata, for example.  

Line 39: Again, the term taeniasis solium. Please, correct this and replace it with the adult of T. solium 

Discussion 

It could be great to discuss the Treg/Th17 and TGFβ in other helminths, including those also located in other tissue.   

White, M. P. J., McManus, C. M., & Maizels, R. M. (2020). Regulatory T-cells in helminth infection: induction, function and therapeutic potential. Immunology, 160(3), 248–260. https://doi.org/10.1111/imm.13190 

Nono, J. K., Lutz, M. B., & Brehm, K. (2020). Expansion of Host Regulatory T Cells by Secreted Products of the Tapeworm Echinococcus multilocularis. Frontiers in immunology, 11, 798. https://doi.org/10.3389/fimmu.2020.00798

Author Response

Dear Reviewer

I am grateful to receive your comments and many thanks for giving me a chance to re-edit this manuscript. and I am most appreciative of your suggestions for my manuscript, which make it more and more perfect. According to the suggestions, I would like to report the revised contents to you:

1. (I cannot find lines 327-328 in connection with response number 4. Could you please highlight this in the Ms?)

Re 1: (Line:325-327) I'm sorry, there may be a misunderstanding here. Now I have revised this sentence again. Are you satisfied with this modification? If not, I will actively revise it. (According to GESA analysis, a total of 21 proteins were identified in the TGF-β signaling pathway and 56 proteins were identified in Th17 cell differentiation, we then performed PPI analysis of the identified 77 proteins.)

2. (Try to write lines 359-360 again, it is redundant when you say liver tissue of mice and then mice model. As a suggestion, you can write something like this: The transcriptional levels of IL-17, TβRI, TβRII, and Smad2/3 in the liver tissue of mice model with Echinococcus multilocularis were significantly increased. However, in the case of prolonged infections, the percentage of CD4+CD25+Foxp3+ Treg cells and serum levels of TGF-β and IL-10 were founded also increased [36].)

Re 2: Thanks for rewriting this sentence, I have written it on the line 366-370 (references [39])

3. (I understand what you want to say but it is not completely correct. You should correct this part for a better understanding of eggs, six-hook larvae, metacestode (Cysticercus cellulosae) and adult (T. Solium), and both C. cellulosae and T. solium can cause disease.)

Re 3: Thanks for rewriting this sentence, I have written it on the line 30-31

4. (Please, delete the host. It is redundant.)

Re 4: Thanks for the suggestion, I've removed " the host" (line 47)

5. (The term taeniasis solium is not normally used and could be weird to read. Please, change this term to taeniasis. Also, it is important to say that the clinical signs and symptoms are undesignable from other gastrointestinal infections, including T. saginata, for example).

Re 5: Thanks for the suggestion, I have replaced "taeniasis solium" with "taeniasis". In addition, I have re-described this sentence (line 34-36), Do you think it is satisfactory? If not, I will actively work on it!

6. (Please, include these papers in the Ms).

Re 6: Many thanks for introducing these 3 articles for me to read, I think these 3 articles have been particularly helpful to me, I have put them in the references [11] (Line 52-56), [35] (Line 357) and [36] (Line 357-361) separately. Please check it out.

The above is my report, and if my manuscript still needs to be revised, I will actively deal with it. Many thanks!

Best wishes
Your Wei He

Reviewer 3 Report

The authors responded to the comments in detail and made suggested changes on the manuscript. I am satisfied with these improvements and I have no other comments. I recommend acceptance of the manuscript.

Author Response

Dear Reviewer

I am grateful to hear from you, many thanks for your affirmation and your support makes me more confident.

Best wishes
Your Wei He